# Changes in Platelet Function in Preterm Newborns with Prematurity Related Morbidities

**DOI:** 10.3390/children9060791

**Published:** 2022-05-27

**Authors:** Irina Franciuc, Elena Matei, Mariana Aschie, Anca Mitroi, Anca Chisoi, Ionut Poinareanu, Nicolae Dobrin, Andreea Georgiana Stoica, Traian Virgiliu Surdu, Mihaela Manea, Sebastian Topliceanu, Georgeta Camelia Cozaru

**Affiliations:** 1Faculty of Medicine, “Ovidius” University of Constanta, 1 Universitatii Street, 900470 Constanta, Romania; irina.franciuc@yahoo.ro (I.F.); ionut_poinareanu@yahoo.com (I.P.); traian@surdu.ro (T.V.S.); 2Neonatology Department, “Sf. Apostol Andrei” Emergency County Hospital, 145 Tomis Blvd., 900591 Constanta, Romania; 3Center for Research and Development of the Morphological and Genetic Studies of Malignant Pathology, Ovidius University of Constanta, CEDMOG, 145 Tomis Blvd., 900591 Constanta, Romania; aschiemariana@yahoo.com (M.A.); ank_mitroi@yahoo.com (A.M.); aka_dobre@yahoo.com (A.C.); dobrinnicolae@gmail.com (N.D.); stoica_andreea_georgiana@yahoo.com (A.G.S.); mihaela.manea@sunmedical.ro (M.M.); topliceanu.sebastian@gmail.com (S.T.); drcozaru@yahoo.com (G.C.C.); 4Clinical Service of Pathology, “Sf. Apostol Andrei” Emergency County Hospital, 145 Tomis Blvd., 900591 Constanta, Romania; 5Internal Medicine-Hematology Department, “Sf. Apostol Andrei” Emergency County Hospital, 145 Tomis Blvd., 900591 Constanta, Romania

**Keywords:** platelet, immature platelets fraction (IPF), respiratory distress syndrome (RDS), intraventricular hemorrhage (IVH), anemia of prematurity (AoP)

## Abstract

Platelet indices represent useful biomarkers to express the thromboembolic status, inflammatory response, and oxidative stress in preterm newborns. Our study presented platelet count and function changes in prematurity-related morbidities such as respiratory distress syndrome, intraventricular bleeding, and anemia of prematurity in preterm newborn cases reported to healthy full-term newborns by flow cytometry and hematological methods. The platelet volume represents the average size of platelets in the blood samples, showing the significantly increased values in preterm newborns compared with healthy full-term newborns due to increasing activated platelet production. Flow cytometric analysis of immature platelet fractions (IPF) made using thiazole orange staining to detect their mRNA content and a glycoprotein (anti-GPIIIa) antibody for platelet gating. CD61-TO expression from premature newborns was significantly lower compared to healthy full-term neonates. Preterm newborn cases with respiratory distress syndrome and a need for respiratory support (RDS+) were characterized by a significantly increased platelet volume and a decreased immature platelet fraction reported in RDS− cases. Evaluating the platelet function in the newborn is difficult because the laboratory methodologies work with small quantities of newborn blood samples. The immature platelet fractions and platelet volume promise to be diagnostic biomarkers for diseases.

## 1. Introduction

Platelets are small discoid cellular particles produced by megakaryocytes that have roles in platelet plug formation, tissue repair, angiogenesis, metastasis, inflammation, and host defense [1,2]. The information about the roles of neonatal platelets in angiogenesis and inflammation is limited. Prematurity-related morbidities have been associated with inflammation [3,4,5]. Intraventricular hemorrhage (IVH) is the most frequent form of early brain injury in preterm newborns. IVH occurrence met in newborns before 32 days of gestational age. The highest risk is for the infant population with respiratory and circulatory failure, developed based on neonatal respiratory distress syndrome (RSD) [6,7]. Several studies show that cerebral sinovenous thrombosis was the frequent cause of symptomatic intraventricular hemorrhage [8]. The dysfunction of platelets is involved in these disorders, but the roles of platelets in the newborn are not yet well defined.

The platelet count is dependent on gestational age, being 150–450 × 10^3^/µL from 22 weeks of gestation age [9,10]. The immature platelets fraction percent (IPF) is increased in healthy newborn circulation [11], but the platelet volume (PV) tends to be comparable to adults. Platelet activation is less effective in the first few days of gestational life, as indicated by the flow cytometric studies, and platelet dysfunctions are common in preterm newborns [12].

Full-term newborns with early gestational age tended to have a decreased platelet number but increased platelet volume [13,14]. PV was increased in preterm newborns with respiratory distress syndrome, suggesting that PV may be a marker of platelet production, consumption, or severity of disorders, thrombosis, and infection [15,16].

In this study, the platelet function was assessed by analyzing the immature platelet fraction (IPF) and mature platelets (MP) percentage by the flow cytometry technique with platelet membranes glycoproteins (GPIIIa) and thiazole orange dual stain, and total platelets number (PLT), platelet volume (PV), platelets index mass (PIM), blood-red cells number (RBC), and hemoglobin value (Hb) parameters made by an automated hematologic analyzer. The clinicopathological aspects and platelet parameters were observed in preterm newborns with prematurity-related morbidities such as respiratory distress syndrome, intraventricular hemorrhage, and anemia of prematurity (AoP) reported in healthy full-term newborns.

## 2. Materials and Methods

### 2.1. Cases Selection

Blood samples of the newborns (their parents signed informed consent forms), were obtained from the Neonatology Department of “Sf. Apostol Andrei” Clinical Emergency County Hospital in Constanta, Romania, collected the following data for all newborns: gender, birth, age, weight, and gestational age. The evaluation of platelet function was made in the Cell Biology Department of the Centre for Research and Development of Morphological and Genetic Studies of Malignant Pathology (CEDMOG), Ovidius University of Constanta, Romania. The control group included the healthy full-term newborns (*n* = 42) from the hospital maternity ward. For experimental premature newborns (*n* = 42) admitted to the Neonatal Intensive Care Unit (NICU), we obtained the following information from their charts: presence (RDS+) or absence of respiratory distress syndrome (RDS−), presence (IVH+) or absence of intraventricular hemorrhage (IVH−), and presence (AoP+) or absence (AoP−) of anemia, designated anemia of prematurity, occurring during their hospital stay.

Intraventricular hemorrhage is bleeding inside or around the ventricles in the brain. There are four grades of IVH, depending on the amount of bleeding: Grade 1. Bleeding occurs just in a small area of the ventricles; Grade 2. Bleeding also occurs inside the ventricles; Grade 3. Ventricles are enlarged by the blood; Grade 4. Bleeding occurs in the brain tissues around the ventricles.

Preterm newborns were classified in the AoP+ category when the hemoglobin concentration after one month at hospital discharge is below 11.5 g/dL. Moreover, the maternal and pregnancy factors were classified into three categories in function of their influence on IPF levels in preterm newborns: (1) Hypertension; (2) Bleeding, cord compression, meconial amniotic fluid; (3) Premature labor.

### 2.2. Reagents and Equipment

Anti-CD61-PE (integrin beta 3, Invitrogen, eBioscience, San Diego, CA, USA) monoclonal antibodies conjugated with phycoerythrin were used to assess platelet expressions of GPIIIa in dual stain with thiazole orange (0.01 mg/mL) at flow cytometry (Attune, Acoustic focusing cytometer, Life Technologies, Carlsbad, CA, USA). Negative control was used CD61-PE (Invitrogen, eBioscience, San Diego, CA, USA). Before the analysis of CD 61-TO, the flow cytometer was set using fluorescent beads (Attune performance tracking beads, Labelling & detection, Life technologies, Carlsbad, CA, USA), with a standard size (1 µL). For data collection, graphics by flow cytometry had used Attune Cytometric Software v.1.2.5, Applied Biosystems, 2010. Pentra XLR automated blood cell analyzer (Horiba Medical, Diamedix, Bucharest, Romania) was used for the total number of platelet (PLT), platelet volume (PV), platelets index mass (PIM), blood-red cells number (RBC), and hemoglobin value (Hb) determinations.

### 2.3. CD 61 Platelet Membrane Glycoproteins and Thiazole Orange Detections by Flow Cytometry Dual Stain

The blood samples were collected into 4 mL tubes with anticoagulant EDTA-K2. All measurements were performed within 120 min after blood withdrawal. Flow cytometry tubes were divided into: (1) samples with CD 61-PE and thiazole orange dual stain; (2) samples with control negative—CD61-PE stain. The flow cytometry tubes introduced 100 µL blood sample, 5 µL of CD 61-PE, and 30 µL of thiazole orange. The contents were gently vortexed and incubated into darkness for 30 min at 37 °C. For each sample, a control tube with 100 µL blood sample and 5 µL of negative control (CD61-PE) was vortexed and incubated in darkness for 30 min at 37 °C. A total of 2 mL of flow cytometry stain buffer was added into each tube and vortexed for one minute before analysis. Mature platelets and immature platelet fractions from blood samples had identified by flow cytometry based on size and specific CD61-TO surface expressions.

### 2.4. Statistical Analysis

We analyzed the birth weight (BW-grams), age (A-days), gestational age (GA-weeks), total platelets number (PLT-103/µL), platelet volume (PV-fL), platelet mass index (PMI), immature platelet fraction (IPF-%), mature platelets (MP-%), blood-red cells number (RBC-10^6^/µL), hemoglobin value (Hb-g/dL), and the obtained results were presented as mean values with standard deviations, being made by SPSS v. 23 software, IBM, 2015. Data were analyzed by the Levene test for homogeneity of the sample variances, an independent *t*-test was used to show the differences between cases, and *p* < 0.05 was considered statistically significant. The Pearson correlations were realized between clinicopathological aspects and platelet parameters on preterm newborns with prematurity-related morbidities. Figure 1 made by Attune Cytometric Software v.1.2.5, Applied Biosystems, 2010.

## 3. Results

### 3.1. Platelet Parameters at Preterm Newborns Reported to Healthy Full-Term Newborns Cases

The descriptive statistic of mean values of clinicopathological aspects and platelet parameters at preterm newborn (experimental-E) and healthy full-term newborn (control-C) cases are presented in Table 1. The birth weight of preterm newborns had significantly lower values than a healthy full-term newborn (E-1433.57 ± 467.77 g vs. C-3316.66 ± 440.67 g, *p* < 0.001). Gestational age had significantly decreased for preterm newborns (30.13 ± 2.98 weeks) compared with healthy full-term newborns (38.76 ± 0.89 weeks).

The platelet volume showed significant differences between groups (E-11.42 ± 1.07 vs. C-9.90 ± 0.62, *p* < 0.001). The immature platelet fraction presented significantly decreased values in preterm newborns reported to healthy full-term newborns (E-11.83 ± 5.70 vs. C-15.78 ± 4.85, *p* < 0.05, Table 1, Figure 1, Appendix A). Preterm newborns presented significantly decreased values of RBC number at birth reported to healthy full-term newborns (E-4.16 ± 0.49 vs. C-4.61 ± 0.51, *p* < 0.05, Table 1) without it (AoP−).

### 3.2. Incidence of Prematurity Related Morbidities in Preterm Newborn Cases

In the experimental group of preterm newborns, the frequencies of clinicopathological characteristics had presented in Table 2. For preterm newborn cases, the most significant incidence of pathologies, 79%, was for respiratory distress syndrome, with the need for respiratory support (RDS+), followed by 71% of cases with anemia of prematurity (AoP+). The intraventricular bleeding disorder (IVH+) was observed in 57% of cases (Table 2).

### 3.3. Platelet Parameters in Preterm Newborns Cases, in the Function of Presence or Absence of Prematurity Related Morbidities

Platelet parameters for preterm newborn cases are shown in Table 3, highlighting the statistically significant differences between them. The platelet volume was significantly higher in newborn cases with RSD and the necessity to use respiratory support (RDS+: 11.58 ± 1.07) compared to those who do not need it (RDS−: 10.50 ± 0.46, *p* < 0.05). Preterm newborns presented significantly lower values of immature platelets fraction in RDS+ reported to RDS− (11.51 ± 4.30 vs. 18.75 ± 9.32, *p* < 0.05, Table 3, Figure 1). RDS+ newborns showed a significantly lower hemoglobin concentration (after one month, at hospital discharge, 11.15 ± 1.20) compared with RDS− newborns (12.00 ± 0.04, *p* < 0.05).

IVH+ newborns had a significantly increased percentage of MP than IVH− newborn (IVH+: 77.77 ± 6.47 vs. IVH−:70.68 ± 3.63, *p* < 0.05, Table 3).

In preterm newborns with AoP+, weight at birth had significantly decreased than AoP− newborns (AoP+: 1263.33 ± 492.72 vs. AoP−:1740.00 ± 207.36, *p* < 0.05). Moreover, IPF was significantly lower for AoP+ neonatal cases than AoP− neonatal cases (AoP+:10.65 ± 2.79 vs. AoP−:16.76 ± 7.77, *p* ≤ 0.05, Table 3, Figure 1, Appendix A).

### 3.4. Correlations between Platelet Parameters and Clinicopathological Aspects in Preterm Newborns

This study observed in the experimental cases of preterm newborns, the significant positive correlations between IPF and clinicopathological characteristics (RDS+: r = 0.398, *p* < 0.05; IVH+: r = 0.376, *p* < 0.05; AoP+: r = 0.390, *p* < 0.05). Moreover, IPF was negatively correlated with platelet volume and mature platelet count (PV: r = −0.389, *p* < 0.05; MP: r = −0.682, *p* < 0.001, Table 4).

The platelet volume presented negative values of correlations with prematurity related morbidities (RDS+: r = −0.698, *p* < 0.001; IVH+: r = −0.529, *p* < 0.01; AoP+: r = −0.549, *p* < 0.01), immature platelet fraction (r = −0.389, *p* < 0.05), and mature platelets count (r = −0.417, *p* < 0.05, Table 5).

In the preterm newborns cohort, the gestational age showed the positive value correlated with weight (r = 0.934, *p* < 0.001), the prematurity related morbidities as RSD+ (r = 0.880, *p* < 0.001), IVH+ (r = 0.755, *p* < 0.001), AoP+ (r = 0.681, *p* < 0.001), negatively correlated with age (r = −0.612, *p* < 0.001), and platelet volume (r = −0.604, *p* = 0.001, Table 6).

The weight of preterm neonates was positively associated with disorders such as respiratory distress syndrome, with needed respiratory support (RDS+: r = 0.818, *p* < 0.001), intraventricular bleeding (IVH+: r = 0.685, *p* < 0.001), anemia of prematurity (AoP+: r = 0.652, *p* < 0.001), and was negatively associated with age (r = −0.559, *p* < 0.01) and platelet volume (r = −0.527, *p* < 0.01, Table 7).

## 4. Discussion

Aggregation involves platelet-to-platelet adhesion being necessary for hemostasis after the initial adhesion of platelets to the injury site. Platelets contain adhesion molecules in the plasma membrane, but also in granules that are important for hemostasis and thrombosis. An important adhesion molecule involved in platelet aggregation is the membrane protein, GPIIb/IIIa complex. This glycoprotein is an integrin receptor present on platelets, both on the plasma membrane and α-granules. In resting platelets, this glycoprotein presented as an inactive form, but their activation by the agonists induces conformational changes of GPIIb/IIIa, being able to bind with soluble plasma fibrinogen [17,18]. The receptor-bound fibrinogen acts as a bridge between two glycoproteins on adjacent platelets [19]. GPIIb/IIIa is the most studied mediator of bridging platelets between them and stabilizing the thrombi [20,21,22]. Platelets have essential roles in procoagulant and fibrinolytic processes because the fibrin binds to aggregated platelets through activated receptor glycoprotein GPIIIa (integrin β3), which helps to stabilize the thrombus [23,24]. Reticulated platelets (RPs) are immature platelets that had newly released from the bone marrow into the circulation, have a high ribonucleic acid content, and are more active in thrombus formation [25,26].

Platelets play a role in a newborn’s inflammatory processes and host defense [6]. Our study presented the platelet count and function changes in disorders such as respiratory distress syndrome, intraventricular bleeding, and anemia of prematurity in preterm newborn cases reported to healthy term newborns by flow cytometry technique and hematological methods.

The platelet volume represents the average size of the platelet in the blood. As reported in other studies, PV showed significantly increased values in preterm newborns compared with healthy full-term newborns [13,14,16]. The difference is due to increased activated platelet production [27,28]. Wiedmeier et al. [29] reported that PV levels between 22 to 42 weeks presented a slight but statistically significant decrease from earlier to later gestation. This trend was observed in our study for platelet volume (11.42 ± 1.07) at earlier gestational age (30.14 ± 2.98) compared with PV (9.90 ± 0.62) at later gestational age (38.66 ± 0.89). Moreover, these authors observed a significant rise in PV over the first two postnatal weeks, a trend also observed in our study. Despite a small number of analyzed samples, our results respect the trend mentioned by Wiedmeier et al. [29] and had included in the reference range between the mean and 5th percentile of this metanalysis.

Flow cytometric analysis of immature platelet fractions were realized using thiazole orange staining to detect their mRNA content and a glycoprotein (anti-GPIIIa) antibody for platelet gating [30]. IPF from premature newborns presented a significantly lower expression of CD61-TO compared to full-term neonates, as reported by scientific literature [31]. The assay for IPF is a non-invasive method of assessing platelet turnover or thrombopoiesis in different types of thrombocytopenia, a simple predictor of bone marrow recovery after transplantation, or used to monitor the response to therapy with growth factors such as thrombopoietin [26].

Preterm newborn cases with respiratory distress syndrome with needed respiratory support (RDS+) were characterized by lower values of platelet count reported in RDS− cases of preterm newborns. Various studies showed that mechanical ventilation reduced platelet counts in newborns with respiratory distress [13,32]. A low platelet count is a well-known biomarker for disease severity.

The percent of immature platelets was a useful biomarker because it was presented that they significantly decreased in preterm newborn cases with pathologies presence (RDS+; AoP+) reported to the absence of these. Recently, attention focused on the platelet role in the pathogenesis of multi-organ failure (MOF), and IPF represents an essential biomarker in MOF [33].

In our study, all newborns have a decrease in circulating RBC levels at birth, and the full-term newborns presented significantly increased values of this reported to preterm newborns. Hb concentrations at birth presented within normal limits for all newborns. Moreover, the preterm newborns with AoP+ with a birth weight higher than 1.0 kg, gestational age below 30 weeks, presented in the first month of age an Hb concentration below 11 g/dL. Authors reported that in healthy full-term newborns, the hemoglobin value rarely decreases below ten g/dL in the first 10–12 weeks of age, being named -physiological anemia of the newborn. In preterm newborns, the blood hemoglobin concentration decreases rapidly, in the first 4–6 weeks of age, to 8 g/dL in neonates with birth weights between 1.0 to 1.5 kg, to 7 g/dL in neonates with birth weights below 1 kg. In many extremely low birth weight (ELBW) newborns, this decreased hemoglobin concentration needs allogeneic RBC transfusions. The anemia of prematurity is not accepted to be physiological [34,35,36]. A principal reason that the hemoglobin concentration is lower in preterm than in full-term newborns is the decreased plasma erythropoietin (EPO) level in response to anemia. Erythroid progenitor cells of newborns are responsive to EPO in vitro, suggesting that inadequate EPO production is a significant cause of neonatal anemia, not unresponsiveness of marrow [34].

In this study, IPF was positively associated with prematurity disorders and negatively correlated with platelet volume and mature platelet count, evaluated as a biomarker for neonatal clinical outcomes in prematurity-related disorders. The role of IPF in preterm and full-term newborns represents a promising clinical biomarker in neonates [37,38]. Saxonhouse et al. [11] reported that among preterm newborns in the first week of life, presented the IPF% was inversely related to gestational age. We did not observe the negative correlations between IPF% and gestational age because the IPF of preterm newborns were analyzed after seven days of life (8.79 ± 8.34). Authors reported no correlation between IPF% and age in newborns with seven days of birth age or older. IPF% determined by the flow cytometry method is suitable for use in full-term and preterm newborns. The preterm newborns presented significantly increased values of the IPF% in the first 2 to 5 days of life, but after that, IPF% had lower values over the first 28 days, being higher in newborns than in adults [11]. IPF% is a new flow cytometric method to measure the young and immature platelet fraction in peripheral blood samples, being a biomarker for thrombopoiesis and adverse cardiovascular outcomes. IPF% is a biomarker for neonatal clinical outcomes pathology such as sepsis [39]. Newborns with hyporegenerative thrombocytopenia (syndromes, small for gestational age, birth asphyxia) presented lower values of IPF% than newborns with consumptive thrombocytopenia (immune-mediated, infection, disseminated intravascular coagulation, necrotizing enterocolitis), to IPF is a biomarker for newborns because these studies clarify the kinetic mechanism which is responsible for thrombocytopenia [40]. The IPF represents the fraction of platelets with small amounts of RNA and rough endoplasmic reticulum [41]. In recent studies, authors reported that IPF% in adult patients presented high sensitivity and specificity, representing a diagnostic biomarker of the infection [42,43,44]. IPF represents an early biomarker for preterm newborns with RSD+, and values of IPF more than 2.9 in the first 24 h after birth may indicate pneumonia [45]. Despite the data deficit in newborns, studies that observe the role of IPF as a sepsis biomarker in adults suggest that this biomarker also provides clinical information about inflammatory activity [45].

We observed an inverse correlation between increased PV levels and the occurrence of morbidities related to prematurity in a preterm newborn cohort, but gestational age (GA) and birth weight (BW) positively correlated with the studied pathologies. Higher PV levels observed in preterm newborns are due to the increased status of activated platelets in the early postnatal weeks, but the inflammatory and thrombotic conditions may vary in function of disease grade. Clinical studies showed that the clinicopathological aspects are associated with increased platelet volume [9,10]. Authors presented that they found in RDS+ newborns higher PV levels, which means that PV could be a predictive biomarker for other encountered diseases in preterm newborns. Authors reported a correlation between PV and morbidities of prematurity such as bronchopulmonary dysplasia (BPD) and intraventricular hemorrhage in a cohort of preterm newborns. PV was significantly higher in newborns with BPD and IVH than in controls on the first day of life. A relationship was found between increased coagulation, reduced fibrinolysis, fibrin deposition in the lungs of RDS+ newborns, and high PV levels [16].

Because the pathophysiologic mechanisms remain confused, increased platelet volume may be associated with inflammatory and thrombotic conditions [46]. PV in preterm newborns informs about hypercoagulative states, increased inflammatory response, and oxidative stress [47]. PV is a platelet activation indicator that is driven by inflammatory processes [48]. Moreover, platelet volume may present an indirect sign of disturbance in platelet activity [49], representing a biomarker of platelet production, and consumption, and may be related to neonatal disease severity, hypoxia, inflammation, and infections [50].

A normal pregnancy is characterized by increased platelet aggregation and decreased circulating platelet count, and PV increases minimally, level with gestation. An increase in platelet consumption in the uteroplacental circulation determinates the reduction in the number of circulating platelets [51].

In our study, the maternal and pregnancy risk factors from the first category, such as hypertension, may influence the IPF levels in preterm newborns by chronic hypoxia and stimulation of fetal erythro/megakaryopoiesis. The bleeding determinates acute hypoxia and a possible medullar response in anemia at birth (Hb < 13 g/dL at birth). The cord compression determinates short-termed hypoxia, meconial amniotic fluid determinates a short-termed/acute fetal distress, possible hypoxia, and premature labor suggests infection factors, but no sepsis developed in our patients. All second and third maternal and pregnancy risk factors present an improbable influence on IPF levels of preterm newborns.

The authors have observed a reduction in platelet count and an elevated platelet size, characteristic of hypertension in pregnancy. The early stages of hypertension during the pregnancy presented an increased level of platelet aggregation, but in severe disease, it is decreased. A higher platelet volume and a lower RBC volume presented in hypertension in pregnancy, indicate disease severity [51,52,53]. Other authors reported higher PV values in diabetic pregnancies, which denote a higher platelet activity that may lead to hypercoagulability being responsible for fatal pregnancy complications [54,55]. Moreover, pregnancy complications incidence increased in women with genetic platelet function disorders. During pregnancy, defects affecting platelets may lead to thrombosis, first-trimester miscarriage, and postpartum hemorrhage [56,57]. The pregnancy induces significant physiologic stress on the pulmonary and cardiovascular systems that may cause respiratory problems. Certain disease states presented to pregnant women, such as amniotic fluid emboli syndrome, were associated with respiratory failure. Moreover, pregnancy-related conditions in common diseases, such as acute respiratory distress syndrome, asthma, pneumonia, and AIDS, must be approached when balancing the fetus’s needs following maternal well-being [58]. Kalaivani [59] reported that maternal anemia increased the risk of preterm births and low birth weight rates. A doubling of the low-birth-weight rate was observed and an increase of 2–3 fold in the perinatal mortality rate when the Hb level is below eight g/dL.

In this study, limitations to developing the utility of platelet function in preterm newborns with prematurity-related morbidities were that a small number of samples recovered from preterm newborns cases and the small quantity of newborn blood used for laboratory methodologies.

Future directions in this research area will be to study the importance of platelet activity in many clinical conditions. Furthermore, immature platelet fractions and platelet volume promise to be critical diagnostic biomarkers for diseases, and the IPF represents the usual biomarker for establishing different types of thrombocytopenia diagnostic.

## 5. Conclusions

Because in our study, the premature newborns with prematurity-related-morbidities had significantly higher values of PV levels reported to healthy full-term newborns, these might be associated with the presence of the inflammatory and oxidative processes. Significantly increased platelet volume values and decreased IPF levels, in RDS+ preterm newborns, reported to RDS− cases may be the biomarkers for disease severity.

Platelet indices, especially the platelet volume and immature platelet fraction, may be useful biomarkers to express the thromboembolic status, inflammatory response, and oxidative stress in preterm newborns.

## Figures and Tables

**Figure 1 children-09-00791-f001:**
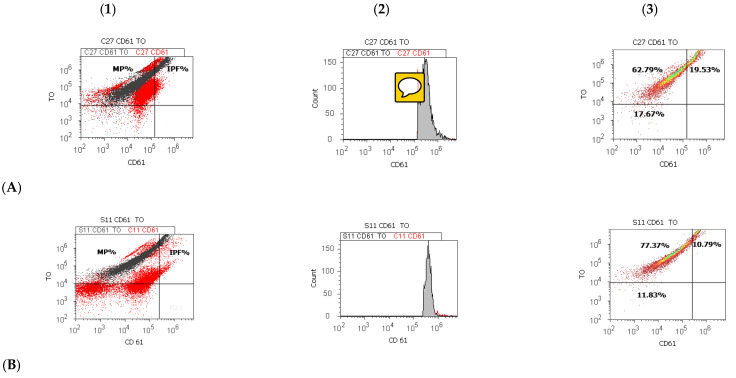
Blood-cell analysis of the immature platelet fraction (IPF) by CD 61 platelet membrane glycoproteins and thiazole orange stain (TO), performed by the Attune Acoustic focusing cytometer, Life Technologies, the USA, equipped with the Attune Cytometric Software v.1.2.5, Applied Biosystems, 2010. (**A**) Healthy full-term newborn with normal IPF; (**B**) Preterm newborn with respiratory distress syndrome with the need for respiratory support (RDS+) and intraventricular bleeding (IVH+) with low IPF; (**C**) Preterm newborn with anemia of prematurity (AoP+) with low IPF; (**D**) Preterm newborn with respiratory distress syndrome with the need for respiratory support (RDS+) with low IPF; (**1**) Dot plot by control extrapolation (CD61) on graphic; (**2**) Histogram plot by IPF% extrapolating on CD61 axis; (**3**) Density plot with quantification of immature platelet fraction (IPF%) and mature platelets (MP%) by highlighting the double-positive stained platelets population represented by IPF.

**Table 1 children-09-00791-t001:** Clinicopathological aspects and platelet parameters in preterm newborns (experimental) and healthy full-term newborns (control) cases.

Nb.	Parameters	Experimental (E) X ± SD	Control (C) X ± SD
1.	Age (days)	8.79 ± 8.34 **	1.13 ± 0.35 **
	*p*-value	**0.004**	
2.	Weight (grams)	1433.57 ± 467.77 **	3316.66 ± 440.67 **
	*p*-value	**0.000**	
3.	Gestational age (weeks)	30.14 ± 2.98 **	38.66 ± 0.89 **
	*p*-value	**0.000**	
4.	Platelets number (10^3^/µL)	325.64 ± 144.33	288.86 ± 107.70
	*p*-value	0.441	
5.	Platelet’s volume (fL)	11.42 ± 1.07 **	9.90 ± 0.62 **
	*p*-value	**0.000**	
6.	Platelets index mass (10^3^/µL × fL)	3724.83 ± 1563.82	2840.00 ± 1019.48
	*p*-value	0.087	
7.	Immature platelet fraction (%)	11.83 ± 3.92 *	15.78 ± 4.91 *
	*p*-value	**0.024**	
8.	Mature platelets (%)	74.70 ± 6.39	70.79 ± 9.68
	*p*-value	0.211	
9.	RBC number at birth (10^6^/µL)	4.16 ± 0.49 *	4.61 ± 0.51 *
	*p*-value	**0.024**	
10.	Hemoglobin value at birth (g/dL)	15.75 ± 0.97	16.44 ± 1.85
	*p*-value	0.220	

**Legend:** X-obtained results mean; SD-standard deviation; ** *p* < 0.01 and * *p* < 0.05 represent significantly statistically differences between controls and experimental samples made by independent *t*-test.

**Table 2 children-09-00791-t002:** Frequencies of clinicopathological aspects in preterm newborn cases.

Nb.	Prematurity Related Morbidities	Experimental Cases Number (%)
1.	Respiratory distress syndrome	RDS+: 33 (79) RDS−: 9 (21)
2.	Intraventricular bleeding	IVH+: 24 (57) IVH−: 18 (43)
3.	Anemia of prematurity (after one month, at hospital discharge)	AoP+: 30 (71) AoP−: 12 (29)

**Legend:** RDS+: preterm newborns with respiratory distress syndrome and need for respiratory support and without it (RDS−); IVH+: preterm newborns with the presence of intraventricular bleeding and absence of this (IVH−); AoP+: preterm newborns with anemia of prematurity and without it (AoP−).

**Table 3 children-09-00791-t003:** Clinicopathological aspects and platelet parameters in preterm newborns cases.

Nb.	Parameters	RDS+ X ± SD	RDS− X ± SD	IVH+ X ± SD	IVH− X ± SD	AoP+ X ± SD	AoP− X ± SD
1.	Age (days)	8.92 ± 9.02	9.00 ± 2.00	11.25 ± 10.38	5.50 ± 2.66	9.56 ± 10.40	7.40 ± 2.40
	*p* values	0.986		0.171		0.662	
2.	Weight (grams)	1397.50 ± 498.19	1650.00 ± 57.73	1256.25 ± 408.09	1670.00 ± 467.29	1263.33 ± 492.72 *	1740.00 ± 207.36 *
	*p* values	0.340		0.103		**0.027**	
3.	Gestational age (weeks)	29.58 ± 2.84 *	33.50 ± 0.57 *	28.75 ± 2.91 *	32.00 ± 2.00 *	29.22 ± 3.07	31.80 ± 2.16
	*p* values	**0.018**		**0.038**		0.125	
4.	Platelets number (10^3^/µL)	320.16 ± 115.89	358.50 ± 283.47	331.12 ± 133.41	318.33 ± 170.67	280.33 ± 69.98	407.20 ± 212.07
	*p* values	0.696		0.877		0.257	
5.	Platelet’s volume (fL)	11.58 ± 1.07 *	10.50 ± 0.46 *	11.61 ± 1.20	11.18 ± 0.92	11.66 ± 1.07	11.00 ± 1.03
	*p* values	**0.015**		0.481		0.283	
6.	Platelets index mass (10^3^/µL x fL)	3701.90 ± 1240.91	3862.45 ± 3142.11	3813.81 ± 1363.61	3606.20 ± 1930.18	3315.03 ± 1026.97	4462.48 ± 2186.47
	*p* values	0.881		0.817		0.200	
7.	Immature platelet fraction (%)	11.51 ± 4.30 *	18.75 ± 9.32 *	11.13 ± 3.22	15.10 ± 7.70	10.65 ± 2.79 *	16.76 ± 7.77 *
	*p* values	**0.047**		0.210		**0.050**	
8.	Mature platelets (%)	75.62 ± 6.41	70.27 ± 3.88	77.77 ± 6.47 *	70.68 ± 3.63 *	76.40 ± 6.89	71.73 ± 4.51
	*p* values	0.078		**0.034**		0.202	
9.	RBC number at birth (10^6^/µL)	4.15 ± 0.53	4.25 ± 0.19	4.18 ± 0.62	4.35 ± 0.17	4.28 ± 0.47	3.96 ± 0.51
	*p* values	0.615		0.529		0.267	
10.	Hemoglobin value at birth (g/dL)	15.73 ± 1.05	15.90 ± 0.04	15.71 ± 1.11	16.01 ± 0.62	15.74 ± 1.22	15.78 ± 0.34
	*p* values	0.597		0.561		0.936	
11.	RBC number after one month, at hospital discharge (10^6^/µL)	3.37 ± 0.43	3.49 ± 0.28	3.53 ± 0.45	3.56 ± 0.44	3.30 ± 0.42 **	3.56 ± 0.38 **
	*p* values	0.552		0.907		**0.002**	
12.	Hemoglobin value after one month, at hospital discharge (g/dL)	11.15 ± 1.20 *	12.00 ± 0.04 *	11.50 ± 0.88	11.55 ± 1.33	10.66 ± 0.95	12.38 ± 0.35
	*p* values	**0.034**		0.934		0.274	

**Legend:** X-obtained results mean; SD-standard deviation; ** *p* < 0.01 and * *p* ≤ 0.05 represent significantly statistically differences between samples made by independent *t*-test; RDS+/RDS−: preterm newborns with respiratory distress syndrome and need for respiratory support (RDS+) and without it (RDS−); IVH+/IVH−: preterm newborns with the presence of intraventricular bleeding (IVH+) and absence of this (IVH−); AoP+/AoP−: preterm newborns with anemia of prematurity (AoP+) and without it (AoP−).

**Table 4 children-09-00791-t004:** Correlations between immature platelet fraction, clinicopathological aspects, and platelet parameters at preterm newborn cases.

Immature Platelets Fraction	Platelet Volume	Need of Respiratory Support	Intraventricular Bleeding	Anemia of Prematurity	Mature Platelets
r	−0.389 *	0.398 *	0.376 *	0.390 *	−0.682 **
*p* values	0.037	0.033	0.045	0.036	0.000

**Legend:** ** *p* < 0.01 and * *p* < 0.05 represent significant statistical differences between samples made by Pearson correlations.

**Table 5 children-09-00791-t005:** Platelet volume correlated with the clinicopathological characteristics and flow cytometry platelet parameters in preterm newborns.

Platelet Volume	Need of Respiratory Support	Intraventricular Bleeding	Anemia of Prematurity	Immature Platelets Fraction	Mature Platelets
r	−0.698 **	−0.529 **	−0.549 **	−0.389 *	−0.417 *
*p* values	0.000	0.003	0.002	0.037	0.024

**Legend:** ** *p* < 0.01 and * *p* < 0.05 represent significant statistical differences between samples made by Pearson correlations.

**Table 6 children-09-00791-t006:** Gestational age correlated with clinicopathological aspects and platelet volume in cases of preterm newborns.

Gestational Age (Weeks)	Age	Weight	Platelet Volume	Need of Respiratory Support	Intraventricular Bleeding	Anemia of Prematurity
r	−0.612 *	0.934 *	−0.604 *	0.880 *	0.755 *	0.681 *
*p* values	0.000	0.000	0.001	0.000	0.000	0.000

**Legend:** * *p* < 0.01 represents significant statistical differences between samples made by Pearson correlations.

**Table 7 children-09-00791-t007:** Preterm newborn weight correlated with prematurity-related disorders and platelet volume in experimental cases.

Weight	Age	Platelet Volume	Need of Respiratory Support	Intraventricular Bleeding	Anemia of Prematurity
r	−0.559 *	−0.527 *	0.818 *	0.685 *	0.652 *
*p* values	0.002	0.003	0.000	0.000	0.000

**Legend:** * *p* < 0.01 represents significant statistical differences between samples made by Pearson correlations.

## Data Availability

Data are contained within the article.

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
