# Peer review of "Changes in Platelet Function in Preterm Newborns with Prematurity Related Morbidities"

_children, 2022, doi:10.3390/children9060791_

Round 1
Reviewer 1 Report
The question investigated by the authors is interesting but major weaknesses preclude manuscript publication of the manuscript in its present form.
There is a lack of methodological details and informations on patients or on clinical parameters allowing to determine their morbidities, such as for example the need for respiratory support or the intraventricular bleeding. In addition, statistical analyses are not robust and even questionable. The Pearson correlation evaluates the linear relationship between two continuous variables ; not clear how maternal and pregnancy risk factors (which are not continuous variables) can be included in this analysis.
References are not up to date and many of them have not been included in the Discussion section (Error! Reference source not found).
Author Response
Response to Reviewer 1
Comment 1: There is a lack of methodological details and informations on patients or on clinical parameters allowing to determine their morbidities, such as for example the need for respiratory support or the intraventricular bleeding. In addition, statistical analyses are not robust and even questionable. The Pearson correlation evaluates the linear relationship between two continuous variables; not clear how maternal and pregnancy risk factors (which are not continuous variables) can be included in this analysis.
Response:
As you suggested, we modified the manuscript and took out of the tables (table 4 and table 5) the Pearson correlations between platelets volume, immature platelets fraction, and maternal and pregnancy risk factors, because the last variable is not continuous.
Comment 2: References are not up to date and many of them have not been included in the Discussion section (Error! Reference source not found).
Response:
As your recommendations, we verified and modified the references from the Discussion section, to be found the references sources from the bibliography, in the manuscript.
Yours faithfully,
Ph.D. Biologist Matei Elena
sogorescuelena@yahoo.com

Reviewer 2 Report
This is an interesting manuscript that is of clinical significance and would be of interest to the scientific community. However, we have the following concerns, only one of which is major, that need to be addressed:
- The manuscript was a little difficult to read, in part, due to the English, but also due to the writing style. This does a dis-service to such an important and significant work that needs to improve.
- Some of the abbreviations are not not defined (for example, GA and BW on bottom of page 9).
- The type of stat tests utilized for various experiments should also be listed under the Statistical analysis section.
- It seems that there are some typos and/or issues related to some of the references that need to be fixed.
- The authors reported differences (on bottom of page 9), between what they found and others in the context of negative correlation between PV and morbidity occurrence related to prematurity. They should elaborate on that.
- The authors should elaborate on the role of IPF as a biomarker in preterm and full-term newborns, and provide additional citations/other studies in support of that argument.
Author Response
Response to Reviewer 2
Comment 1: The manuscript was a little difficult to read, in part, due to the English, but also due to the writing style. This does a dis-service to such an important and significant work that needs to improve.
Response:
As you recommended, we modified a little the writing style in the manuscript, so that the content can be easily understood.
Comment 2: Some of the abbreviations are not not defined (for example, GA and BW on bottom of page 9).
Response:
As you suggested, we verified and defined the abbreviations in the manuscript, inclusive of page 9.
“In a preterm newborn cohort, we observed a negative correlation between PV and the occurrence of morbidities related to prematurity, but gestational age (GA) and birth weight (BW) were positively correlated with the presence of the studied pathologies.”
Comment 3: The type of stat tests utilized for various experiments should also be listed under the Statistical analysis section.
Response:
All types of statistical tests are mentioned in the Statistical analysis section.
“We analyzed the birth weight (BW-grams), age (A-days), gestational age (GA-weeks), total platelets number (PLT –103/µl), platelet volume (PV – fL), platelet mass index (PMI), immature platelet fraction (IPF- %), mature platelets (MP-%), blood-red cells number (RBC-106/µl), hemoglobin value (Hb-g/dL), and the obtained results were presented as means values with standard deviations, being made by SPSS v. 23 software, IBM, 2015. Data were analyzed by the Levene test for homogeneity of the sample variances, an independent T-test was used to show the differences between cases, and P<0.05 was considered statistically significant. The Pearson correlations were made between clinicopathological aspects and platelet parameters on preterm newborns with prematurity-related morbidities”.
Comment 4: It seems that there are some typos and/or issues related to some of the references that need to be fixed.
Response:
As your recommendations, we verified and modified the references from the Discussion section, to be found the references sources from the bibliography, in the manuscript.
Comment 5: The authors reported differences (on bottom of page 9), between what they found and others in the context of negative correlation between PV and morbidity occurrence related to prematurity. They should elaborate on that.
Response:
As you suggested, we elaborated on the context of the observed correlations between the increased platelet volume and incidence of the pathologies and modified the manuscript such as:
“In a preterm newborn cohort, we observed an inverse correlation between increased PV levels and the occurrence of morbidities related to prematurity, but gestational age (GA) and birth weight (BW) were positively correlated with the presence of the studied pathologies. Higher PV levels observed in preterm newborns are due to the increased status of activated platelets in the early postnatal weeks, but the inflammatory and thrombotic conditions may vary in function of disease grade. Results of clinical studies have shown that the clinicopathological aspects are associated with the increase in platelet volume [9, 10]. Authors presented that, they found at RDS+ newborns higher PV levels, mean that PV could be a predictive biomarker for other encountered diseases in preterm newborns. Authors reported a correlation observed between PV and the occurrence of various morbidities of prematurity such as bronchopulmonary dysplasia (BPD), and intraventricular hemorrhage in a cohort of preterm newborns. PV was significantly higher in newborns with BPD, and IVH, in comparison with controls on the first day of life. A relationship was found between increased coagulation and/or reduced fibrinolysis and fibrin deposition in lungs of RDS+ newborns and high PV levels [16].“
Comment 6: The authors should elaborate on the role of IPF as a biomarker in preterm and full-term newborns, and provide additional citations/other studies in support of that argument. Congratulations for the team work and for the effort made in carrying out the project.
Response:
At your recommendations, we elaborated on the role of IPF as a biomarker in preterm and full-term newborns and modified the manuscript. Also, the references were added to the manuscript.
“IPF% determined by the flow cytometry method is suitable for use in full-term and preterm newborns. The preterm newborns presented significantly increased values of the IPF% in the first 2 to 5 days of life, but after that IPF% had lower values over the first 28 days, being higher in newborns than in adults [11]. IPF% is a new flow cytometric method to measure the young and immature platelet fraction in peripheral blood samples, being a biomarker for thrombopoiesis and negative cardiovascular outcomes. IPF% is a biomarker for neonatal clinical outcomes pathology such as sepsis [39]. Newborns with hyporegenerative thrombocytopenias (syndromes, small for gestational age, birth asphyxia) presented lower values of IPF% than newborns with consumptive thrombocytopenias (immune-mediated, infection, disseminated intravascular coagulation, necrotizing enterocolitis), so as to IPF is a biomarker for newborns because these studies clarify the kinetic mechanism which is responsible for thrombocytopenias [40]. The IPF represents the fraction of platelets with small amounts of RNA and rough endoplasmic reticulum [41]. In recent studies, authors reported that IPF% in adult patients presented high sensitivity and specificity, being a diagnostic biomarker of the infection [42-44]. IPF represents an early biomarker for preterm newborns with RSD+, and values of IPF more than 2.9 in the first 24 hours after birth may be indicative of pneumonia [45]. Despite the data deficit in newborns, studies that observe the role of IPF as a sepsis biomarker in adults suggest that this biomarker also provides clinical information about inflammatory activity [45].”
Yours faithfully,
Ph.D. Biologist Matei Elena
sogorescuelena@yahoo.com

Round 2
Reviewer 1 Report
I have no additional comments.
Author Response
Dear Reviewer 1
Thank you, for the comments and recommendations. The corrections have been implemented in the manuscript.
As your suggestion, I revised the manuscript and improved the English language and style.
Yours faithfully,
Ph.D. Biologist Matei Elena

Reviewer 2 Report
In general I am satisfied by the revisions, albeit there is still some room for improvement in the English.
Author Response
Dear Reviewer 2
Thank you, for the comments and recommendations. The corrections have been implemented in the manuscript.
As your suggestion, I revised the manuscript and improved the English language and style.
Yours faithfully,
Ph.D. Biologist Matei Elena

This manuscript is a resubmission of an earlier submission. The following is a list of the peer review reports and author responses from that submission.
Round 1
Reviewer 1 Report
In this original article entitled “Changes in Platelet Function in Preterm Newborns with Prematurity Related Morbidities” Franciuc et al. aimed to investigate platelet function in preterm infants with different prematurity-related morbidities in comparison to healthy full-term newborns. Importantly, preterm neonates have an increased risk for the development of neurological impairment (e.g. IVH) and other systemic complications causing increased risk of neonatal morbidity and mortality. Early diagnosis is very important and non-invasive biomarker research in this field is still a ‘hot-topic’ to prevent further severe complications.
The authors demonstrated that mean platelet volume (MPV) and immature platelet fraction (IPF, reticulated platelets) was significantly lower in preterm infants vs. controls. Additionally, IPF and MPV showed moderate or strong association with prematurity related morbidities based on Pearson correlations.
Overall, the manuscript has merits and fits to the scope of the Journal. Despite of these interesting findings, some improvements should be addressed before final decision.
Major comments:
- In this study the Authors investigate preterm infants with anemia, however, the Reviewer did not find any results about it. What were the RBC count and hemoglobin concentrations of the preterm neonates and controls? Please insert some data about these issues (e.g. Results, Table 1).
- Platelet function was investigated via measuring IPF by flow cytometry. Did the Authors measure platelet activation markers (e.g. surface P-selectin or soluble P-selectin concentration) or platelet aggregation? If so, please insert some data into the Results and the Discussion about the platelet activation status of these patients.
- One of the main finding is that reticulated platelet:mature platelet ratio was significantly lower in premature infants with different complications vs. healthy controls. Figure 1 shows some representative flow cytometry dot plots, where the size and location of the gates are not consistent, therefore the differences in IPF percentage can be imprecise. It is somewhat confusing, please clarify what was the strategy of gating. Did the Authors use negative control (isotype control or PBS etc.) without TO to set the gates for analyzing double positive platelets?
Minor comments:
- English language of the manuscript needs to be revised and improved.
- In Table 1 and Table 3 please indicate the exact p-values where appropriate.
- In the Results section (2.2 paragraph) or in the Materials and Methods (4.1) please characterize the grades of IVH.
- Please include the permit number of the ethical approval.
- It is recommended to use higher resolution in case of Figure 1.
- In Table 1 paired t-test was used for comparison of experimental and control groups, however, in this case unpaired t-test is more relevant. Please, recalculate these p-values.
Reviewer 2 Report
In their manuscript 'Changes in platelet function in preterm newborns with prematurity related morbidities' Franciuc et al, analyzed blood obtained from 42 preterm newborns and compared it with blood from healthy full-term newborns in terms of platelet number, volume, mass index, immature platelet fraction % and mature platelets %.
The title is misleading. The authors did not perform any assays related to platelet function such as aggregometry, platelet adhesion assays or stimuli-dependent platelet activation assays. Instead, they showed significantly increased platelet volume and decreased reticulated platelets. However, a study from Wiedmeier et al (J perinat 2009) including 47000 subjects, which was not taken into consideration in this manuscript, showed that there are no differences in platelet volume between preterm and full-term newborns. This arises by the sample size which is insufficient in this case. An older study that was again not taken into consideration is the one from Arad et al Am J Perinatol 1986 that is in line with Wiedmeier et al. The authors observed a decreased percentage of immature platelet fraction in the preterm newborns. However, Saxonhouse et al., (J Pediatr Hematol/Oncol 2004) found a negative correlation between immature platelets and gestational age.
Next, they compared prematurity-related morbidities such as respiratory distress syndrome, intraventricular bleeding and anemia of prematurity. Within the preterm newborns they the ones that required respiratory support to those that they did not. However, the sample size is too small to draw conclusions (n=9 in the RDS- group). In this small subject number, they analyzed again platelet parameters. They did not however take into consideration pregnancy and delivery complications that might impact on platelet function (Trudinger et al, Obstet Gynecol 2003).
Figure 1 shows the blood-cell analysis of the immature platelet fraction. The authors need to improve the quality of the figure.
The authors found correlations between immature platelets fraction, platelet volume and clinicopathological aspects. Again, this is surprising taking into account the publication of WWiedmeier et al that analyzed more than 47000 newborns.